# Functional network modules overlap and are linked to interindividual connectome differences during human brain development

**Tianyuan Lei**[1,2,3,4], **Xuhong Liao**[5]\*, **Xinyuan Liang**[2,3,4], **Lianglong Sun**[2,3,4], **Mingrui Xia**[2,3,4], **Yunman Xia**[2,3,4], **Tengda Zhao**[2,3,4], **Xiaodan Chen**[2,3,4], **Weiwei Men**[6,7], **Yanpei Wang**[2], **Leilei Ma**[2], **Ningyu Liu**[2], **Jing Lu**[2], **Gai Zhao**[2], **Yuyin Ding**[2], **Yao Deng**[2], **Jiali Wang**[2], **Rui Chen**[2], **Haibo Zhang**[2], **Shuping Tan**[8], **Jia-Hong Gao**[6,7,9], **Shaozheng Qin**[2,3,4,10], **Sha Tao**[2], **Qi Dong**[2], **Yong He**[2,3,4,10]\*

1 Department of Psychiatry, Beijing Children's Hospital, Capital Medical University, National Center for Children's Health, Beijing, China, 2 State Key Laboratory of Cognitive Neuroscience and Learning, Beijing Normal University, Beijing, China, 3 Beijing Key Laboratory of Brain Imaging and Connectomics, Beijing Normal University, Beijing, China, 4 IDG/McGovern Institute for Brain Research, Beijing Normal University, Beijing, China, 5 School of Systems Science, Beijing Normal University, Beijing, China, 6 Center for MRI Research, Academy for Advanced Interdisciplinary Studies, Peking University, Beijing, China, 7 Beijing City Key Laboratory for Medical Physics and Engineering, Institute of Heavy Ion Physics, School of Physics, Peking University, Beijing, China, 8 Psychiatry Research Center, Beijing Huilongguan Hospital, Peking University Huilongguan Clinical College, Beijing, China, 9 IDG/McGovern Institute for Brain Research, Peking University, Beijing, China, 10 Chinese Institute for Brain Research, Beijing, China

\* liaoxuhong@bnu.edu.cn (XL); yong.he@bnu.edu.cn (YH)

**Data Availability Statement:** All code and data used in the analyses are available in the Zenodo

## Abstract

The modular structure of functional connectomes in the human brain undergoes substantial reorganization during development. However, previous studies have implicitly assumed that each region participates in one single module, ignoring the potential spatial overlap between modules. How the overlapping functional modules develop and whether this development is related to gray and white matter features remain unknown. Using longitudinal multimodal structural, functional, and diffusion MRI data from 305 children (aged 6 to 14 years), we investigated the maturation of overlapping modules of functional networks and further revealed their structural associations. An edge-centric network model was used to identify the overlapping modules, and the nodal overlap in module affiliations was quantified using the entropy measure. We showed a regionally heterogeneous spatial topography of the overlapping extent of brain nodes in module affiliations in children, with higher entropy (i.e., more module involvement) in the ventral attention, somatomotor, and subcortical regions and lower entropy (i.e., less module involvement) in the visual and default-mode regions. The overlapping modules developed in a linear, spatially dissociable manner, with decreased entropy (i.e., decreased module involvement) in the dorsomedial prefrontal cortex, ventral prefrontal cortex, and putamen and increased entropy (i.e., increased module involvement) in the parietal lobules and lateral prefrontal cortex. The overlapping modular patterns captured individual brain maturity as characterized by chronological age and were predicted by integrating gray matter morphology and white matter microstructural properties. Our findings highlight the maturation of overlapping functional modules and their

database (https://osf.io/qfcyu/; doi: 10.17605/OSF.IO/QFCYU) and on GitHub (https://github.com/helab207/Development-of-the-overlapping-modular-structure-in-human-brain-functional-networks).

**Funding:** The study was supported by the grant from the National Key R&D Program of China (https://service.most.gov.cn/index/) (grant 2018YFA0701402 to Y.H.), the National Natural Science Foundation of China (https://www.nsfc.gov.cn/) (grants 82021004, 81620108016 to Y.H., grants 81971690, 11835003 to X.L., grants 31221003, 31521063 to Q.D., and grant 81801783 to T.Z.), the Beijing Brain Initiative of the Beijing Municipal Science & Technology Commission (https://kw.beijing.gov.cn/) (grant Z181100001518003 to S.T.), and the Tang Scholar Award of Beijing Normal University (http://www.tangfoundation.org.cn/) (grant to X.L.). The funders had no role in study design, data collection and analysis, decision to publish, or preparation of the manuscript.

**Competing interests:** The authors have declared that no competing interests exist.

**Abbreviations:** AIC, Akaike information criterion; FDR, false discovery rate; FOV, field of view; GQI, generalized q-sampling imaging; HARDI, high angular resolution diffusion imaging; mFD, mean framewise displacement; MPRAGE, magnetization prepared rapid acquisition gradient-echo; rsfMRI, resting-state fMRI; SVR, support vector regression.

structural substrates, thereby advancing our understanding of the principles of connectome development.

## Introduction

Childhood and adolescence is a period of transition from infancy to adulthood, which is critical for the maturation and improvement of motor, cognitive, emotional, and social functions [1,2]. During this period, the brain undergoes progressive and regressive maturation in its microscopic and macroscopic anatomy, such as increased myelination [3,4], synaptic pruning [4–6], and cortical thinning [4,7–9]. From the perspective of function, remarkable reconfigurations have also been observed for task-evoked regional activity [10,11] and task-free spontaneous activity [12]. These structural and functional changes are critical for improvements in children's cognitive and behavioral performance [10,11,13]. Notably, the periods of childhood and adolescence are also critical windows for the onset of many psychiatric disorders [14]. Exploring brain developmental principles in children and adolescents would provide insights into understanding not only cognitive and behavioral growth but also neurodevelopmental disorders.

Over the past 2 decades, neuroimaging-based connectomics has provided a valuable framework for investigating the developmental principles of brain function [15–17]. The functional modular structure, which is characterized by dense within-module connections and sparse between-module connections, has attracted great attention [18–21]. The modular structure of the brain is particularly important for global network communications, as it can facilitate efficient information segregation and integration with low wiring costs [20,22]. Several studies have reported age-related changes in modular organization with development [15,16,23–25]. Specifically, functional modular organization is already present in fetuses [26] and neonates [27,28]; in this configuration, the modules in the primary cortex show an adult-like topography, and the modules in the association cortex are far from mature. The modular architecture further undergoes an elaborate reconfiguration from childhood to adulthood [15,16,23,24]. Its spatial layout shifts from an anatomical proximity to a spatially distributed and functionally related configuration [25]. These changes have been linked to the development of individual cognition and behavior, such as cognitive control [24] and general cognitive function [23].

Despite this substantial progress, previous connectome development studies have focused primarily on modular structure without spatial overlap (i.e., hard assignment), implicitly assuming that each brain node belongs to 1 single functional module. This assumption may be problematic because the modular structures of real-world networks, such as cooperation networks in social systems and protein networks in nature, generally show overlapping properties [29,30]. This overlapping modular framework in complex networks provides important insights into the potential diverse functional roles of nodes in the network. Several recent functional network studies have also reported overlapping modular organization in adults [31–37], indicating that brain regions are not restricted to one specific module. In particular, the overlap between functional modules is spatially heterogeneous [32–35], with higher overlapping regions being crucial for intermodule communication [38] and global network efficiency [33]. The presence of overlapping modules is also related to spatial heterogeneity in functional diversity and neurocognitive flexibility [32,33,39], and changes with aging [39,40]. However, the network growth principle of the overlapping modular organization in the functional connectome and its association with gray and white matter features remain unknown.

To fill these knowledge gaps, we investigated the development of the overlapping modular architecture of functional connectomes during childhood and adolescence and examined their associations with gray and white matter features using a large longitudinal multimodal neuroimaging data set from 305 typically developing children (aged 6 to 14 years, 491 scans in total) [41–43]. Specifically, we investigated the development of the overlapping functional modules using resting-state fMRI (rsfMRI) data and further revealed the underlying structural substrates of the overlapping modules using structural and diffusion MRI data. In the present study, we identified overlapping functional modules using an edge-centric module detection approach [44], which makes no assumptions about the probabilistic membership of nodes. This approach provides an intuitive way to assign nodes to various communities based on the module affiliations of their edges (referred to as nodal soft partitioning) [45]. The overlapping extent of each brain node in the module affiliations was quantified with an entropy measure [34]. We then examined the development in the overlapping extent of children's brain networks at the global, system, and nodal levels, and further assessed whether the spatial pattern of the modular overlap can predict brain maturity. Finally, we investigated the potential structural substrates involved in the development of overlapping functional modules.

## Results

We leveraged longitudinal rsfMRI data from 305 children (aged 6 to14 y, 491 scans), including 3 repeated scans from 47 children, 2 repeated scans from 92 children, and 1 scan from 166 children (Fig 1A). For comparison purposes, we also included cross-sectional rsfMRI data from a group of healthy adults (*n* = 61, aged 18 to 29 y). Both children and adults were scanned using the same scanner with identical protocols. All MR images used here underwent strict quality control (see Materials and methods). We identified the overlapping modules in individual- and group-level functional networks using an edge-centric module detection algorithm [30,44] (Fig 1B). Briefly, we first constructed a traditional functional network comprising nodal regions and interregional connections (i.e., edges). Then, we constructed a weighted

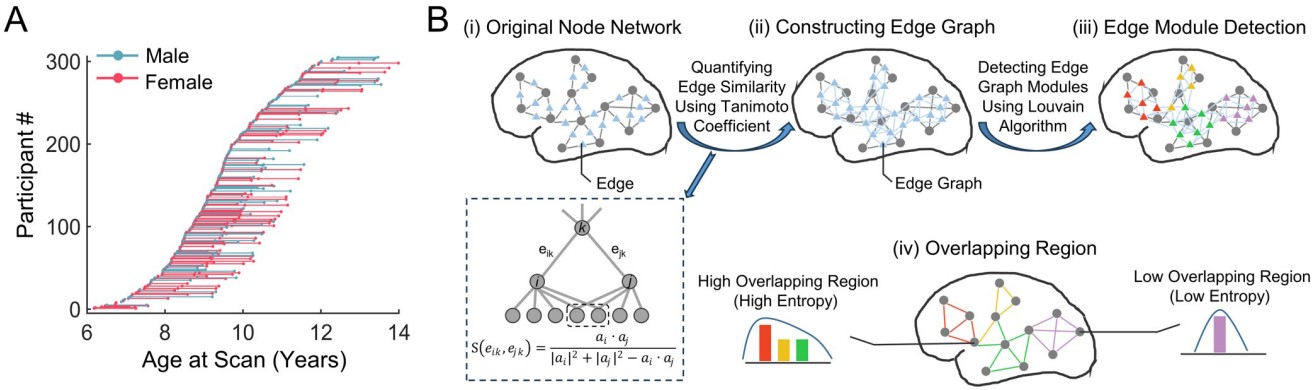

**Fig 1. Data information and schematic diagram of the overlapping modular architecture based on the edge-centric module detection. (A)** Age distribution of longitudinal rsfMRI scans of children. **(B)** (i) Traditional brain functional connectivity network. In this network, each node denotes a brain region of interest, and each link denotes the interregional functional connectivity. (ii) Edge graph corresponding to a given functional network. In this graph, each node denotes an edge in the functional network, and each link is defined as the similarity between edges in the connectivity profiles using Tanimoto coefficient [30,46]. For 2 given edges $e_{ik}$ and $e_{jk}$ that share a common node $k$, the interedge similarity was estimated as the similarity of connectivity profiles between node $i$ and node $j$, wherein $a_i$ represents the modified connectivity profile of node $i$, $a_i \cdot a_j$ represents the dot product of 2 vectors $a_i$ and $a_j$, and $|a_i|^2$ denotes the sum of the squared weights of all connections of node $i$. (iii) Edge-centric module detection. Each edge is assigned to a specific module based on the Louvain algorithm [47]. (iv) Definition of regional module overlap. Each nodal region was assigned to one or more modules due to the diverse module affiliations of its edges. A measure of entropy was employed to quantify the extent of module overlap of each brain node by measuring the distribution of the module affiliations of its edges [34]. rsfMRI, resting-state fMRI.

edge-based brain graph that represented the similarity of connectivity profiles between edges (see Materials and methods). Finally, we identified module affiliations of each node according to the module assignments of its edges in the corresponding edge graph. A measure of entropy was used to estimate the extent of modular overlap for each node by quantifying the distribution of module affiliations of the edges attached to this node [34] (Fig 1B).

## Spatial topography of the overlapping functional modules in children and adults

We first identified the overlapping functional modules in healthy young adults, which serves as a reference for exploring the development of the overlapping modules in children. We found 7 modules in the weighted edge-based brain graph of the adult group (Fig 2A) and further showed the corresponding topographic distribution of each module (Fig 2B and 2C). These functional modules showed substantial spatial overlap, as characterized by 73% of the nodal regions belonging to 2 or more modules (Fig 2B). Module I was mainly located in the medial and lateral prefrontal and parietal cortex, and lateral temporal cortex; module II was mainly located in the primary motor and somatosensory cortices; module III was mainly located in the insula, supramarginal gyrus, superior temporal gyrus, and paracentral lobule; module IV was mainly located in the cingulate gyrus and the subcortical area; module V was located in the visual cortex and the superior parietal lobule; module VI was located in the middle frontal gyrus and superior parietal lobule; and module VII was primarily located in the temporal pole, hippocampus, and amygdala (Fig 2C).

We further divided all the children's rsfMRI scans into 8 subgroups with a one-year interval, and the adult group was set as the ninth subgroup for comparison. We identified the overlapping modular architecture for each subgroup of children based on the rsfMRI data. The modules in each child subgroup were matched with those in the adult subgroup. In general, most functional modules in the child subgroups showed high spatial similarity with those in the adult subgroup (Pearson's correlation $r$s: mean ± SD = 0.75 ± 0.02, range: 0.25 to 0.95) (Fig 2D). Initial inspection revealed that the spatial similarity with the adult group tended to increase with age for all modules, except modules V and VI. For each module, the spatial distribution of nodes among prior functional systems [48,49] was largely consistent across all child subgroups and the adult cohort (Fig 2E). Module I mainly involved the default-mode and frontoparietal systems; module II mainly involved the somatomotor, ventral attention, and dorsal attention systems; module III mainly involved the somatomotor and ventral attention systems; module IV mainly involved the ventral attention, frontoparietal, and default-mode systems, and the subcortical area; module V mainly involved the visual and dorsal attention systems; module VI mainly involved the visual, somatomotor, dorsal attention, and ventral attention systems; and module VII mainly involved the limbic and default-mode systems and the subcortical area. Interestingly, we found that the nodal regions contained in each module mainly belong to the functional systems located in the adjacent hierarchy [50,51], regardless of the subgroup.

## Development of the overlapping functional modules during childhood and adolescence

We employed the mixed effects model [53,54] to quantify the longitudinal changes in the overlapping modular structure during childhood and adolescence. At the global level, the number of modules in the edge-based brain graphs significantly decreased with age (linear model, $t = -2.09$, $p = 0.037$), and the modularity tended to increase with age (linear model, $t = 1.89$, $p = 0.059$) (Fig 3A). The global entropy (i.e., mean nodal entropy across the brain) did not

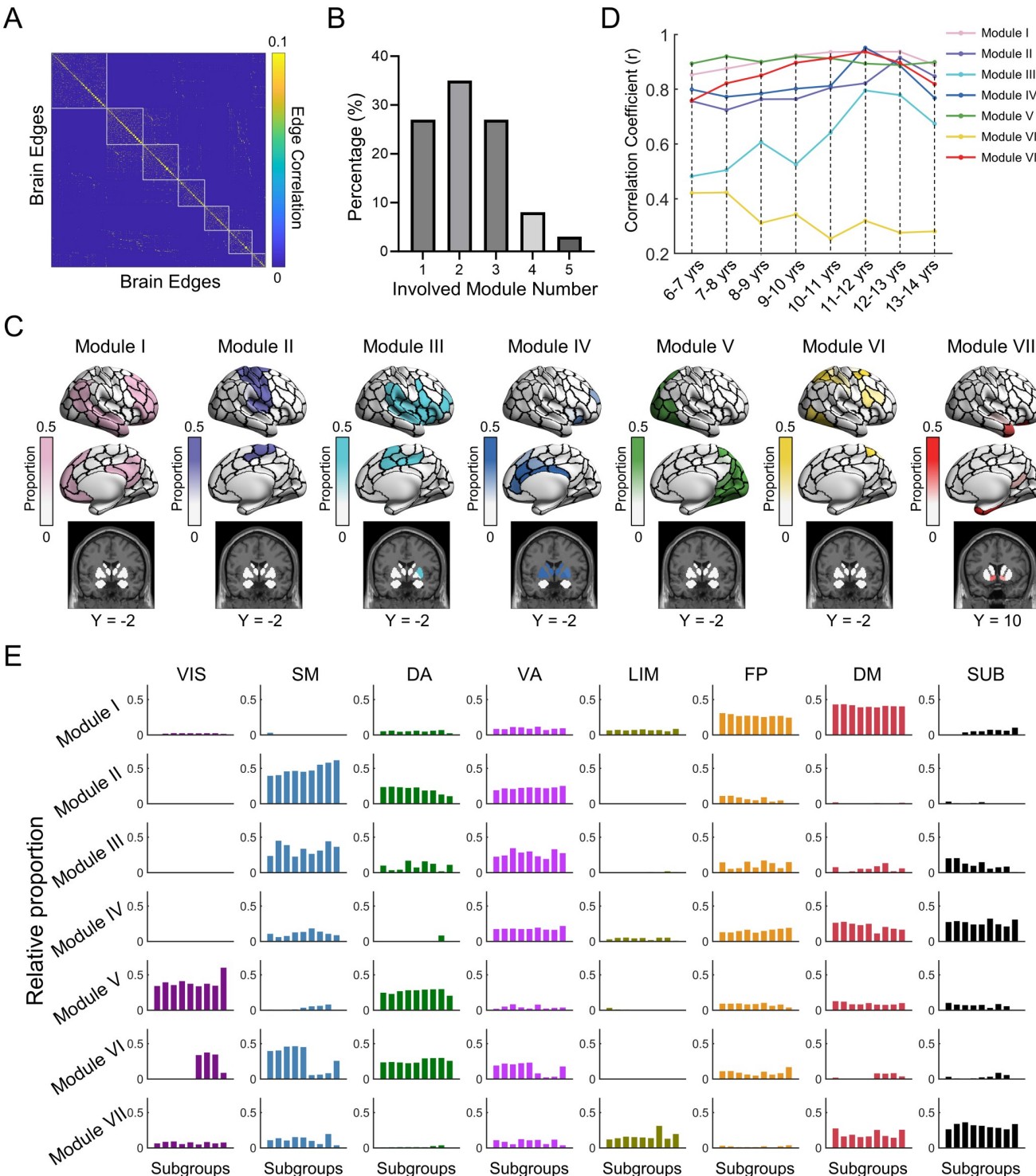

**Fig 2. Overlapping modular architecture of group-level brain functional networks in the adult cohort and the child subgroups. (A)** Modular organization in the weighted edge graph in adults. The edges were sorted according to their module affiliations. Each element denotes the interedge similarity in their connectivity profiles. **(B)** Distribution of the involved module number across brain nodes. Notably, 27% of the nodes belonged to one module, and 73% of the nodes belonged to 2 or more modules. **(C)** Topographic distributions of 7 functional modules. For each module, nodal values represent the proportion of edges assigned to that module. **(D)** Spatial similarity of functional module maps between the child subgroups and the adult group. Each line represents the age-dependent similarity for a particular module. **(E)** System-dependent spatial distributions of functional modules. For each functional module of each subgroup, we calculated the percentage of nodes distributed in eight systems, including 7 functional systems [48] and the subcortical area [49]. Given a prior system, the bar chart shows the percentage of nodes located in this system for 8 child subgroups with a one-year interval

(i.e., 6–7 yrs, 7–8 yrs, 8–9 yrs, 9–10 yrs, 10–11yrs, 11–12 yrs, 12–13 yrs, and 13–14 yrs) and the adult cohort. In (C) and (D), cortical data were mapped on the brain surface using BrainNet Viewer software [52]. The data underlying this figure can be found at https://osf.io/qfcyu/. VIS, visual; SM, somatomotor; DA, dorsal attention; VA, ventral attention; LIM, limbic; FP, frontoparietal; DM, default-mode; SUB, subcortical; yrs, years.

show significant correlation with age ($t = -1.54$, $p = 0.12$). At the regional level, the spatial topography of the nodal overlap (i.e., entropy) of every child subgroup was highly similar to that of the adult subgroup (Pearson's correlation $r$s ranged from 0.74 to 0.91). Specifically, for each age subgroup, regions with higher levels of module overlap (i.e., higher entropy) were located mainly in the insula, supramarginal gyrus, inferior frontal gyrus, somatosensory cortex, anterior cingulate gyrus, superior temporal gyrus, and subcortical regions (e.g., putamen), and regions with lower levels of module overlap (i.e., lower entropy) were located mainly in the visual cortex, angular gyrus, and posterior cingulate gyrus (Fig 3B). Statistical analysis revealed that nodal entropy in 7 nodal regions showed significant linear changes with age (FDR-corrected $p < 0.05$, Fig 3C). These regions showed dissociable developmental patterns, with significant increases in entropy (i.e., increased module involvement) mainly in the superior and inferior parietal lobules and lateral prefrontal cortex and significant decreases in entropy (i.e., decreased module involvement) mainly in the ventral prefrontal cortex, dorsal medial prefrontal cortex, and putamen.

At the system level, two-way repeated analysis of variance (ANOVA) revealed that nodal entropy exhibited a system-dependent distribution (system effect: $F(7,77) = 117.30$, $p < 0.0001$; age subgroup effect: $F(7,77) = 1.21$, $p = 0.31$; interaction effect: $F(49,539) = 1.44$, $p = 0.03$) (Fig 4A). Specifically, the ventral attention and somatomotor systems showed greater module overlap (i.e., higher entropy), while the visual and default-mode systems showed lower module overlap (i.e., lower entropy). Quantitative analysis revealed that the entropy of the dorsal attention system significantly increased with age (linear model, $t = 2.44$, $p = 0.015$), while the entropy in the subcortical area significantly decreased with age (linear model, $t = -3.13$, $p = 0.0019$, Fig 4B).

## Association of cognitive functions with developmental changes in overlapping modules

We then performed a meta-analysis using the NeuroSynth database [55] to explore the potential cognitive significance associated with developmental changes in the overlapping modules. The statistical map of the developmental changes in nodal entropy was divided into 10 bins with decreasing age-related $T$ values. We found that the nodal regions that increased with age (bins of 0%–10% and 10%–20%), such as the lateral prefrontal cortex, superior parietal lobule, and inferior parietal lobule, were mainly involved in the cognitive terms "motor imagery," "visual perception," "spatial," and "eye movements" (Fig 5). Nodal regions showing decreases with age (bins of 80%–90% and 90%–100%), such as the ventral prefrontal cortex, the dorsal medial prefrontal cortex, and the putamen, were mainly associated with the cognitive terms of "speech," "word form," "sound," and "auditory."

## Predicting chronological age from spatial topography of nodal overlap

We investigated whether the spatial topography of nodal overlap in the network modules could be used to predict individual chronological age. Linear support vector regression (SVR) was used with 10-fold cross-validation (Fig 6A). To avoid the possibility of data leakage that could occur by including scans from the same children in both the training and test sets, a total of 305 rsfMRI scans from independent subjects were selected for use in the prediction

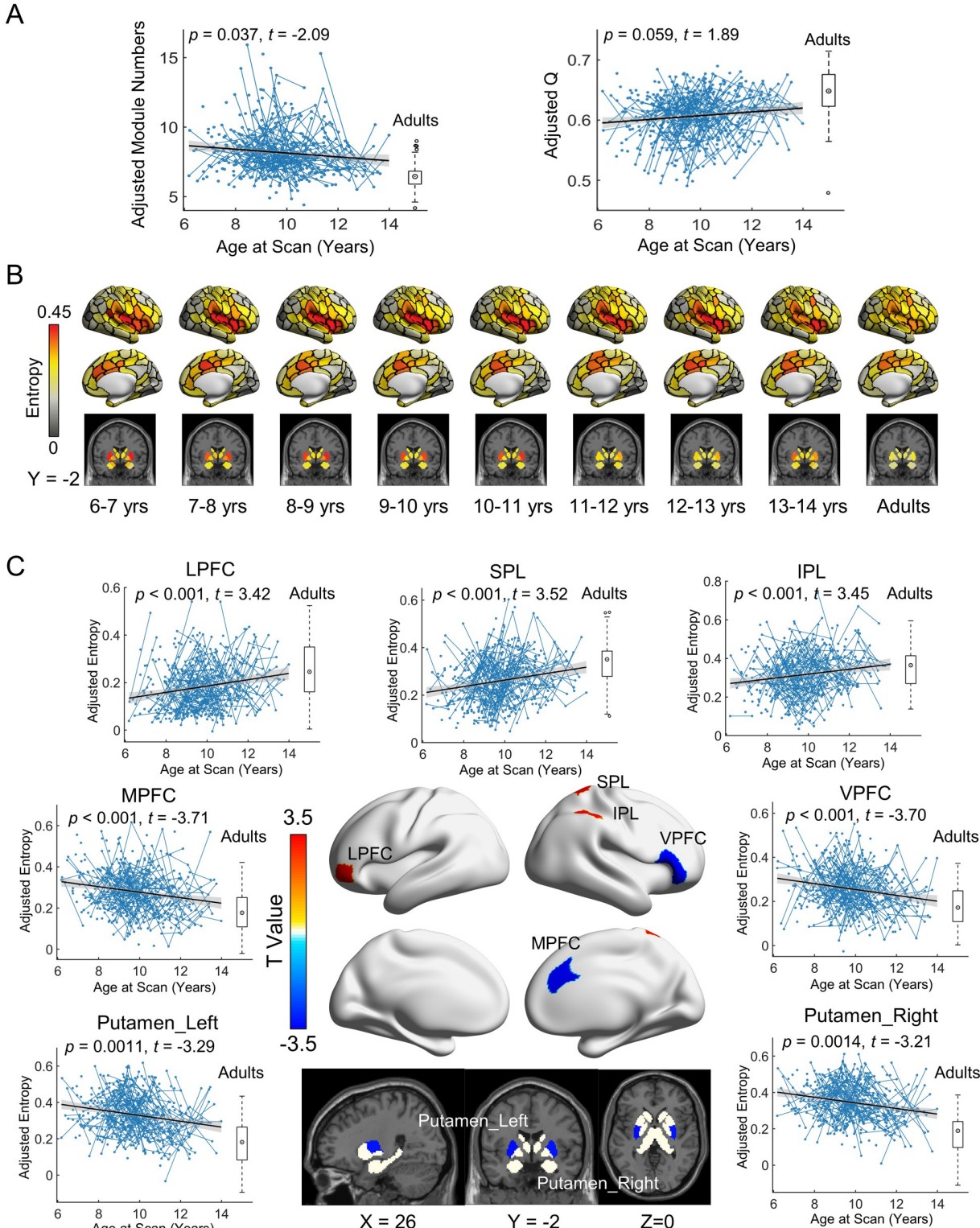

**Fig 3. Longitudinal development of overlapping functional modules at the global and nodal levels. (A)** Left: Age effect on the number of modules. Right: Age effect on modularity in the edge graph. **(B)** Spatial patterns of functional module overlap (i.e., nodal entropy) across the brain for each child subgroup and for the adult group. **(C)** Spatial distribution of regions showing significant developmental changes in nodal entropy. Age effects are displayed in terms of $t$ values (FDR-corrected $p < 0.05$, corresponding to uncorrected $p < 0.0014$). In (A) and (C), the boxplot represents the distribution of the adult group for reference. The blue lines connecting scattered points represent longitudinal scans of

the same child. The adjusted value denotes the measure of interest corrected for sex, head motion, and random age effects. The data underlying this figure can be found at https://osf.io/qfcyu/. yrs, years; LPFC, lateral prefrontal cortex; SPL, superior parietal lobule; IPL, inferior parietal lobule; MPFC, medial prefrontal cortex; VPFC, ventral prefrontal cortex.

analysis. For a given individual with longitudinal scans, only one scan was randomly selected from the individual. The age prediction analysis was repeated 1,000 times with random scan selection. We found that the spatial patterns of nodal entropy significantly predicted individual chronological age ($r = 0.14 \sim 0.52$, all $p_{perm}$s $< 0.05$; Fig 6B). The significance level of the prediction accuracy was assessed with permutation tests ($n = 10,000$) in which the original age was shuffled across children. Regions with high contributions were primarily located in the dorsal attention, ventral attention, and default-mode systems (Fig 6C). Furthermore, we found that the nodal contribution weights showed a significant positive correlation with the

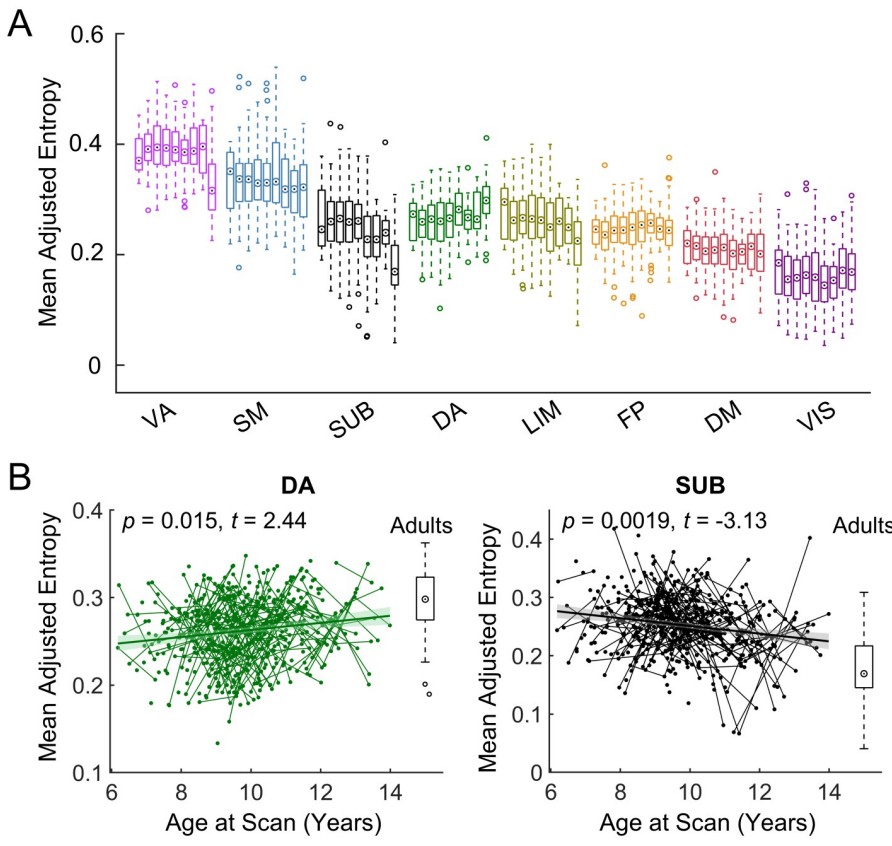

**Fig 4. Longitudinal development of the overlapping functional modules at the system level. (A)** Distribution of nodal entropy within each functional system for each child subgroup and for the adult group. Given a prior system, the bar chart shows the distribution of the average entropy of this system across individuals for 8 child subgroups with a one-year interval (i.e., 6–7 yrs, 7–8 yrs, 8–9 yrs, 9–10 yrs, 10–11yrs, 11–12 yrs, 12–13 yrs, and 13–14 yrs) and the adult cohort. Here, the circle with a dot denotes the median, and the box denotes the interquartile range. **(B)** Two functional systems showing significant developmental changes in functional module overlap (i.e., nodal entropy). The boxplot represents the distribution of the adult group for reference. Short lines connecting scattered points represent longitudinal scans of the same child. The adjusted value denotes the measure of interest corrected for sex, head motion, and random age effects. The data underlying this figure can be found at https://osf.io/qfcyu/. VIS, visual; SM, somatomotor; DA, dorsal attention; VA, ventral attention; LIM, limbic; FP, frontoparietal; DM, default-mode; SUB, subcortical; yrs, year.

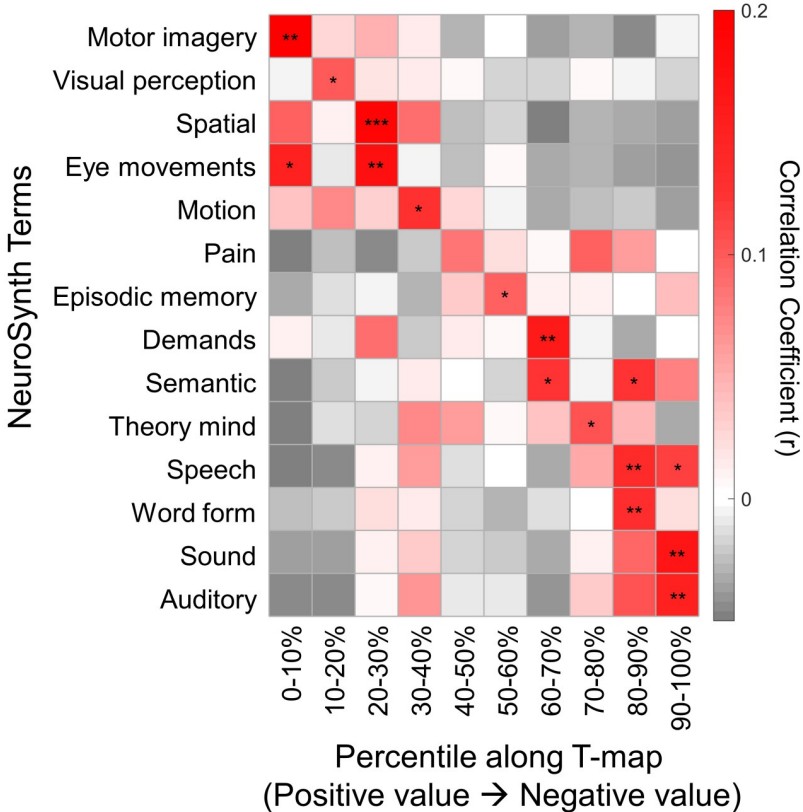

**Fig 5. Cognitive function decoding of brain regions.** Nodal regions were sorted in descending order according to developmental changes in the functional module overlap. The results were obtained based on the information in the NeuroSynth meta-analytic database [55]. The significance level of the spatial similarity was assessed using permutation tests ($n = 10,000$) that corrected for spatial autocorrelation [56]. The data underlying this figure can be found at https://osf.io/qfcyu/. *, $p_{perm} < 0.05$; **, $p_{perm} < 0.01$; ***, $p_{perm} < 0.001$.

developmental changes in nodal entropy in terms of age-related $t$ values (Pearson's correlation $r = 0.48$, $p_{perm} < 0.0001$; Fig 6D). The significance level of the spatial similarity was assessed using permutation tests ($n = 10,000$) to correct for spatial autocorrelation [56]. This result suggests that brain regions showing age-related changes play a crucial role in predicting chronological age.

## Predicting individual spatial topography of overlapping functional modules from structural brain features

We finally investigated whether structural features were related to the overlapping modules in children. In this analysis, we included 446 high-quality rsfMRI, structural and diffusion MRI scans from 279 children (aged 6 to 14 years, F/M = 138/141), with 3 repeated scans from 42 children, 2 repeated scans from 83 children, and 1 scan from 154 children. For each brain node of each child, we obtained 6 structural features, including 5 morphological measurements (cortical volume, thickness, curvature, folding index, and surface area) using structural MRI data and 1 white matter microstructural measure (fractional anisotropy (FA) strength) using diffusion MRI data. We initially examined the spatial similarity between nodal entropy in the overlapping functional modules and each structural feature within each scan using the univariate Pearson's correlation. Cortical thickness showed a positive correlation with nodal

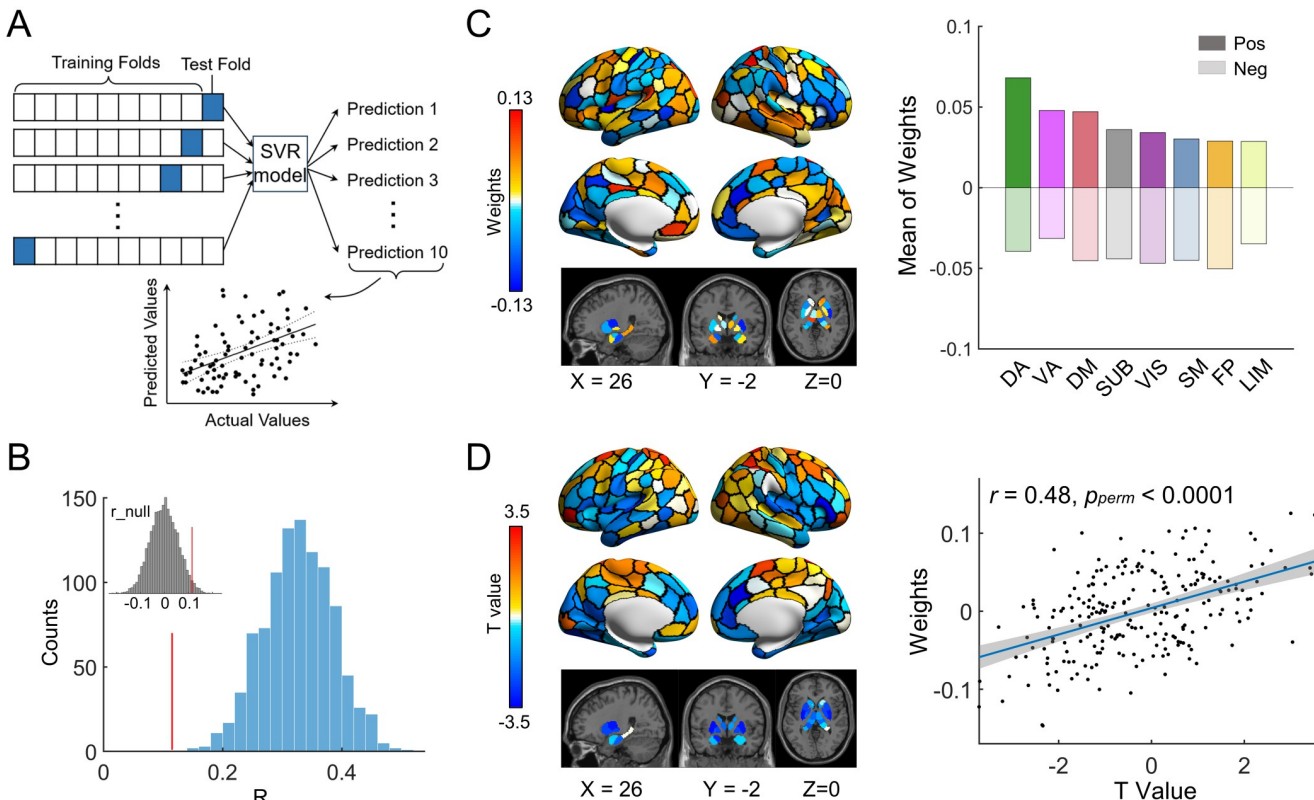

**Fig 6. Age prediction based on spatial patterns of nodal overlap. (A)** Schematic representation of the SVR prediction model based on 10-fold cross-validation. **(B)** Frequency polygon for age prediction accuracy using the 10-fold SVR model for 1,000 times of randomly selected samples (blue histogram). For each time of random sampling, rsfMRI scans were randomly selected from 305 independent subjects. The gray frequency polygon in the inset displays the null distribution of prediction accuracy based on the permutation tests ($n = 10,000$) by randomly sampling the scans and shuffling the original ages across the scans. The red line in the blue histogram indicates the significant level ($p < 0.05$) derived from the null distribution. **(C)** Spatial distributions of the nodal contribution in the prediction model. Left: Contribution weight at the regional level. Right: Contribution weight at the system level. The nodal contribution weights were obtained by averaging the contribution weights across 1,000 times of randomly selected samples. Positive and negative weights were separately averaged within each system. **(D)** Left: Spatial pattern of age effect of on nodal overlap in terms of $t$ values. Right: Nodal contribution weight shows a significant positive correlation with the development of nodal overlap based on Pearson's correlation analysis. The significance level of the similarity was assessed using the permutation tests ($n = 10,000$) to correct for spatial autocorrelation [56]. The data underlying this figure can be found at https://osf.io/qfcyu/. Pos, positive; Neg, negative; VIS, visual; SM, somatomotor; DA, dorsal attention; VA, ventral attention; LIM, limbic; FP, frontoparietal; DM, default-mode; SUB, subcortical; SVR, support vector regression.

entropy, while the other structural features exhibited negative correlations (S1 Fig). Next, the individual spatial pattern of nodal entropy was predicted by integrating all structural brain features using the SVR model. We found that the structural features significantly predicted the individual spatial pattern of nodal entropy for 95% of the scans (424/446), with a significance level of $p_{perm} < 0.05$. The prediction accuracy varied across scans (mean ± SD: 0.35 ± 0.10), while the maximum prediction accuracy reached 0.59 (Fig 7A). The prediction contributions varied across structural features, with cortical thickness showing the greatest contribution (Fig 7B). Fig 7C and 7D show the prediction accuracy and contributing features for a representative child's scan.

## Validation results

We first validated the procedures used for the estimation of nodal overlap in the functional module affiliations. Specifically, we re-estimated the nodal overlap in module affiliations in the

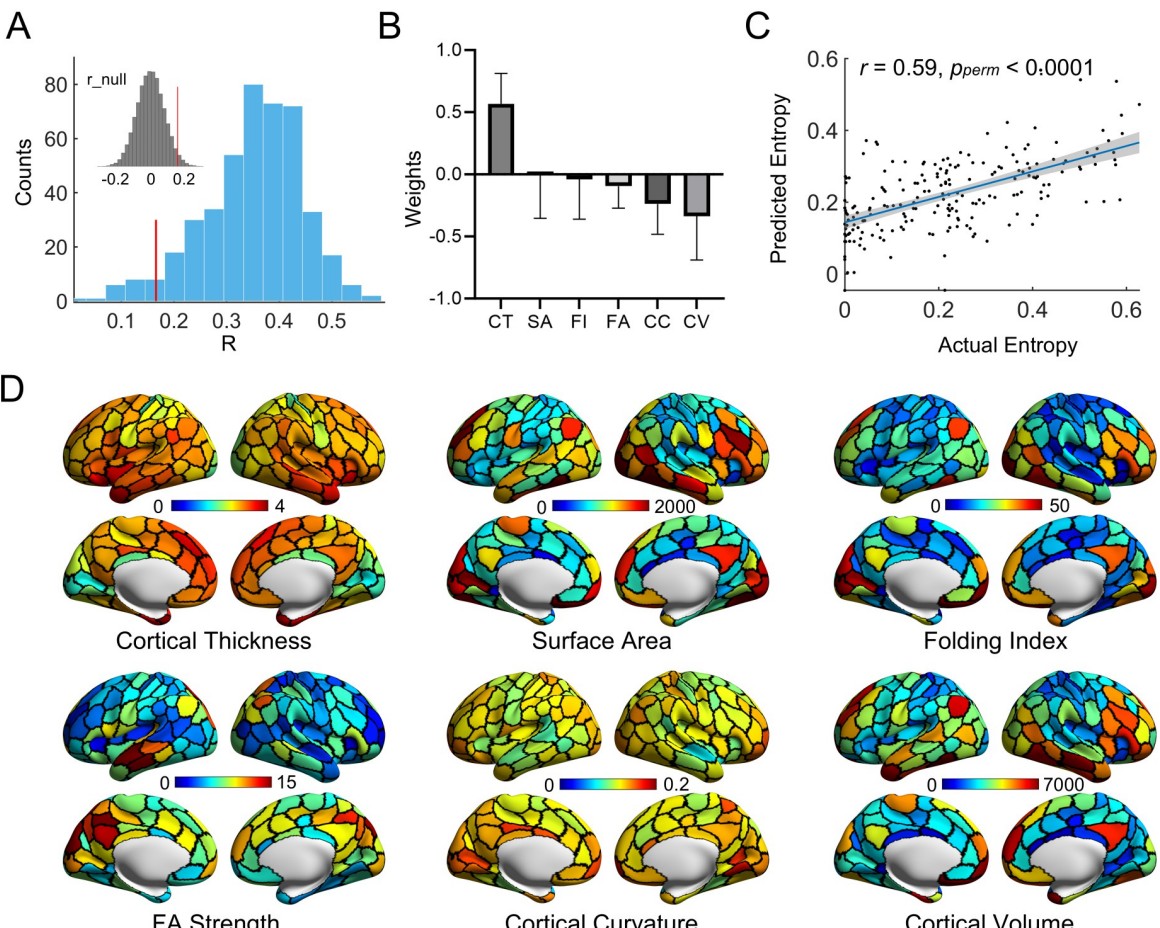

**Fig 7. Prediction of individual spatial patterns of nodal module overlap from structural brain features in children. (A)** Frequency polygon for prediction accuracy of all rsfMRI scans. The inset in the upper left corner denotes the null distribution of the prediction accuracy based on permutation tests. To assess the statistical significance of the prediction accuracy, we generated a null distribution of accuracy using permutation tests by shuffling the original entropy values across nodes for each scan 100 times, thus leading to 44,600 (446 scans × 100 times) permutation instances in total. The red line denotes the 95% significance level in the null distribution. **(B)** Prediction contribution of different structural features in the prediction model of all rsfMRI scans. This histogram displays the mean contribution across subjects for different anatomical features, with each bar representing the mean accompanied by its standard deviation. **(C)** Accuracy of nodal entropy prediction for a representative child's scan. **(D)** Spatial distributions of anatomical features for a representative child's scan. In (C) and (D), the representative scan was selected as the scan that showed the highest prediction accuracy for the individual map of nodal module overlap (i.e., nodal entropy). The data underlying this figure can be found at https://osf.io/qfcyu/. CT, cortical thickness; SA, surface area; FI, folding index; FA, fractional anisotropy; CC, cortical curvature; CV, cortical volume.

adult group by considering the influence of several network construction and analysis strategies, including (i) spatial resolution of the parcellation (i.e., Schaefer-100) (S2 Fig); (ii) network thresholding strategies (i.e., 10% and 20%) (S3A and S3B Fig); (iii) module detection algorithms (S3C Fig); and (iv) measures for nodal overlap estimation (i.e., involved number) (S3D Fig). We found that the spatial pattern of nodal entropy remained almost unchanged in different cases, suggesting the robustness of the topography of the overlapping modular architecture.

Subsequently, the development of nodal entropy was validated by employing different network thresholding strategies (i.e., 10%, 20%, and 30%) (S4 Fig) and the generalized additive mixed model (GAMM) for estimating age effects (S5 Fig). The spatial patterns of age effects on

nodal entropy remained highly similar to the main results, with spatial correlations ranging from 0.70 to 0.98. Furthermore, a more stringent head motion control strategy was implemented by excluding rsfMRI scans with mFD > 0.2 mm (retaining 355 scans from 233 children). The age effect map exhibited a high degree of spatial correlation ($r = 0.89$, $p_{perm} <$ 0.0001) with the main results (S6 Fig). These sensitive analyses indicate that the age effects on nodal entropy were robust across methodological variations.

## Discussion

This study employs a large longitudinal multimodal MRI data set and an edge-centric module detection approach to elucidate the spatially dissociable development of overlapping functional modular organization and their structural associations during childhood and adolescence. Specifically, we observed the presence of an adult-like, spatially inhomogeneous nodal modular overlap in children. The nodal modular overlap (i.e., nodal entropy) developed with remarkable spatial heterogeneity, with significant decreases in the ventral and medial prefrontal cortex and putamen and significant increases in the superior and inferior parietal lobules and the lateral prefrontal cortex. These changes suggest an age-related reorganization in intermodule integration. The spatial patterns of nodal overlap were found to predict individual chronological age and were associated with morphological and white matter features, particularly cortical thickness. These findings highlight the gradual maturation of the overlapping modular architecture and its potential structural substrates during childhood and adolescence, providing novel insights into the developmental rules of the human brain.

The functional network exhibits an intricate community structure with multi-scale organization in both adults [57] and youths [58], and its spatial topography varies across individuals [21,36,37]. In this study, we used the novel framework of overlapping modular architectures, which provides an intuitive representation of the functional interactions between modules and the multiple roles of brain regions [59,60] and deepens the understanding of the organizational principles of the human brain [31–35]. Compared to previous studies on overlapping modules in adults [31–35] and the elderly [40], our research focuses on the development of overlapping functional modular organization during childhood and adolescence. We found that each functional module comprised nodal regions involved in at least 2 prior functional systems, regardless of the age subgroup. These systems contained in the same functional module have usually been found to be located in adjacent functional gradients [50]. This may be partially attributable to the fact that regions locating in proximate functional hierarchy tend to show strong functional interactions [58]. Taken together, these findings indicate that the overlapping modular architecture captures intersystem interactions at similar hierarchical levels. Interestingly, most of the prior functional systems were involved in 3 or more overlapping modules, except for the visual and limbic systems, suggesting diverse functional roles or functional differentiation within each brain system. Specifically, the default-mode regions were found to be involved in three overlapping modules (i.e., modules I, IV, and VII), which is in line with the previous identification of 3 subsystems within the default-mode system in both children [42] and adults [61]. From a dynamic perspective, functional modular organization spontaneously reconfigures on a short time scale (e.g., seconds) with regions switching among modules [21]. The observed spatial overlap of functional modules may be a summary of the dynamic reorganization at a long-term scale. However, the relationship between the overlapping and dynamic modular architectures warrants further investigation.

We observed that the topographical distribution of the module overlap in children and adolescents showed an adult-like spatially inhomogeneous pattern. This suggests that the overlapping modular architecture is taking shape in children and adolescents. High nodal overlap was

primarily located in the ventral attention, somatomotor, and medial and inferior frontal regions, which is consistent with the findings of a recent study in adults [34]. The high nodal overlap of the ventral attention system may be due to its involvement in multiple general domain categories, as revealed in a meta-analytic study [59], including cognition (e.g., attention), perception (e.g., somesthesis), and action domains (e.g., imagination). During a movie watching task, the module overlap level of the ventral attention system increased significantly [34], further indicating that the module overlap of the ventral attention system may capture regional involvement in task transitions. The high nodal overlap in the sensorimotor regions may indicate a high level of between-module integration of this system, which aligns with previous findings that sensorimotor networks exhibit greater between-network coupling in comparison to association cortices [58,62]. The high nodal entropy in these regions may be attributed, in part, to their tendency to exhibit nested multi-scale organization, as demonstrated in previous studies [63]. Despite the highly specialized functions of sensorimotor regions [50,60], the potential functional significance of the high overlap in these regions warrants further investigation.

Some recent studies have identified fuzzy network affiliations of brain regions in children using the clustering or template matching approaches, but ignoring their developmental changes at this stage [36,64]. Here, we found the maturation in the nodal modular overlap (i.e., nodal entropy) is regionally heterogeneous. The decreased nodal overlap of subcortical areas reflects the gradually decreased involvement of the subcortical system in modules III and V, which are related to somatomotor, attention, and visual functions (Fig 2E). A recent study reported that the strength of cortico-subcortical functional connectivity varies with age, with increasing connections between subcortical and association regions and weakening connections between subcortical and primary regions [65]. The age-related decreased overlap of the subcortical regions suggests that the enhanced functional segregation between the subcortical area and the primary system may be more dominant. In the prefrontal cortex, we observed different age-related changes in nodal overlap for the lateral prefrontal regions and the dorsal medial prefrontal regions. The divergent changes might be due to the fact that these 2 regions belong to different sub-modules of the frontoparietal system [66]. The lateral prefrontal region strongly connects with the default-mode system that is primarily involved in the regulation of introspective processes, and the dorsal medial prefrontal region strongly connects with the dorsal attention system that is mainly involved in the regulation of visuospatial perceptual attention [66]. The significant prediction of an individual's chronological age based on nodal overlap patterns further demonstrated substantial changes in the overlapping modular architecture during childhood and adolescence. Of note, during the adult lifespan, the age-dependent overlapping functional modules are associated with fluid cognition [40]. Further exploration is warranted regarding the potential relationship between the maturation of the overlapping functional modules and individual cognitive development.

The functional activity of the human brain is supported and sculpted by the underlying anatomical structure [67]. During childhood and adolescence, both gray matter and white matter of the brain undergo elaborate reconfiguration. The cortical surface area gradually reaches a peak, and the gray matter volume of the whole brain decreases [9]. Moreover, the white matter volume continues to increase [9,68]. Here, we found that these developing anatomical features of gray matter and white matter could be used to significantly predict the spatial patterns of functional module overlap at the individual level. Of all the measures considered, cortical thickness was found to make the largest contribution. Previous studies have documented that cortical thickness shows a remarkable thinning during childhood and adolescence [9,69,70]. The cortical thinning may be related to the microstructural changes within the cortical layers, such as synaptic pruning and intracortical myelination, as well as mechanical forces that affect

cortical morphology [4]. Interestingly, a previous study has shown that the coordinated development of cortical thickness during adolescence shows a similar pattern of functional connectivity [71]. These results provide a link between the overlapping functional modular architecture and structural features in the brain.

Previous studies on overlapping modules in the human brain have primarily used node-centric methods [32,33,35,39], usually assuming probabilistic membership or maximal clique for nodes, which struggle to capture intricate overlapping configuration [72]. In this study, we employed an edge-centric algorithm [30,44] to identify the overlapping modular architecture for brain nodes, offering an intuitive and detailed representation of multiple module affiliations of nodes [30,34]. A recent study suggests that the overlapping modular architecture revealed by the edge-centric approach can be explained by surrogate interedge similarity derived from the node-level functional connectivity [73]. Further research is needed to explore the differences between these 2 approaches. Of note, identifying the overlapping modular architecture using the current edge-centric method faces several challenges. First, uncertainties remain regarding whether and how the functional correlation matrix is thresholded to construct a stable node-level functional network [74,75]. Second, this study only considers the relationships between edge pairs that share common nodes, neglecting potential indirect relationships between edges. Higher-order relationships between edges could be considered in future studies.

Several issues need to be further considered. First, inner-scan head motion introduces a spatially inhomogeneous bias in functional connectivity estimation, which is age dependent [76]. To reduce the potential influence of head motion, we excluded rsfMRI scans with extensive head motion and further included 24 head parameters, global brain signals, and "bad" time points during nuisance regression [77,78]. The individual head motion parameters (i.e., mFD) were also included in the mixed effects model when assessing the age effects [77,79]. When performing a more stringent head motion control, the developmental effects of nodal entropy remained almost unchanged. These results suggest that the head motion has a limited effect on our findings. Nevertheless, it would be valuable to further refine head motion correction methods to mitigate the residual influence of head motion on functional network development. Second, this study used a large longitudinal data set of children and adolescents, but the age range was limited to school-age children. Previous studies have already identified changes in the overlapping modular properties of the human brain during aging [39,40]. Future studies should extend the period of interest by adding more data sets to chart a lifespan trajectory. Third, the structure–function association was established using an SVR prediction analysis. However, the spatial distribution of the overlapping modules in functional networks does not align consistently with that in structural networks [44,80]. Further work is needed to elucidate how structural brain networks shape functional overlapping modules during development by establishing computational network models, such as communication models [81] and large-scale dynamic modeling [82]. Finally, we explored the typical developmental changes in the overlapping modules in children and adolescents. Since adolescence is the most common period for the onset of mental disorders [14], the overlapping modular organization may be altered in individuals with neurodevelopmental disorders, such as autism spectrum disorders or attention deficit hyperactivity disorder. Investigating the overlapping modular organization in these disorders may deepen insights into the pathological mechanism underlying atypical development.

## Methods

### Participants

A longitudinal multimodal MRI data set of 360 healthy children was obtained from the Children School Functions and Brain Development Project (Beijing Cohort) [41–43]. All

participants were cognitively normal, did not use psychotropic medication, and had no history of severe traumatic brain injury. These children underwent longitudinal rsfMRI, structural and diffusion MRI scans at intervals of approximately 1 year. We performed strict quality control on the rsfMRI data and excluded scans with field map errors, excessive head motion (see "Data preprocessing"), excessive "bad" time points, or T1 artifacts. Finally, 491 scans of 305 children (aged 6 to 14 years, F/M = 143/162) remained, including 3 scans from 47 children, 2 scans from 92 children, and 1 scan from 166 children. Notably, 45 rsfMRI scans were further excluded from the subsequent structural association analysis due to the poor quality of the corresponding dMRI data (see "Data preprocessing"). The remained 446 rsfMRI scans from 279 children (aged 6 to 14 years, F/M = 138/141) were used in the structure–function association analysis, including 3 scans from 42 children, 2 scans from 83 children, and 1 scan from 154 children. In addition, we employed rsfMRI scans of 61 healthy young adults (aged 18 to 29 years, F/M = 37/24) for comparison. All participants or their parents/guardians provided written informed consent. This study was designed and performed according to the principles of the Helsinki Declaration and was approved by the Ethics Committee of Beijing Normal University (approval No. IRB_A_0004_2019001).

## Imaging acquisition

Multimodal magnetic resonance images were acquired on a 3T Siemens Prisma scanner with a 64-channel head coil at the Center for Magnetic Resonance Imaging Research at Peking University. Both children and adults underwent multimodal scanning using the following protocols.

(*i*) *Functional MRI*. Resting-state scans were acquired using an echo-planar imaging sequence: repetition time (TR) = 2,000 ms, echo time (TE) = 30 ms, flip angle = 90˚, field of view (FOV) = 224 × 224 mm$^2$, acquisition matrix = 64 × 64, slice number = 33, and slice thickness/gap = 3.5/0.7 mm. All participants were asked to fixate on a bright crosshair displayed in the center of the scanner screen. The total duration of the rsfMRI scans was 8 min (i.e., 240 volumes).

(*ii*) *Field maps for functional MRI*. The scans were acquired using a 2D dual gradient-echo sequence: TR = 400 ms, TE1 = 4.92 ms, TE2 = 7.38 ms, flip angle = 60˚, FOV = 224 × 224 mm$^2$, acquisition matrix = 64 × 64, slice number = 33, and slice thickness/gap = 3.5/0.7 mm.

(*iii*) *T1-weighted structural MRI*. The scans were acquired using a sagittal 3D magnetization prepared rapid acquisition gradient-echo (MPRAGE) sequence: TR = 2,530 ms, TE = 2.98 ms, flip angle = 7˚, FOV = 256 × 224 mm$^2$, acquisition matrix = 256 × 224, inversion time = 1,100 ms, slice number = 192, slice thickness = 1 mm, and bandwidth = 240 Hz/Px.

(*iv*) *Diffusion MRI*. The scans were acquired using a single-shell high angular resolution diffusion imaging (HARDI) sequence [83,84]: TR = 7,500 ms, TE = 64 ms, flip angle = 90˚, FOV = 224 × 224 mm$^2$, acquisition matrix = 112 × 112, slice number = 70, slice thickness = 2 mm, and bandwidth = 2,030 Hz/Px. The complete sequence consisted of 64 diffusion-weighted directions (b-value = 1,000 s/mm$^2$) and 10 non-diffusion-weighted directions (b-value = 0 s/mm$^2$). The use of multiple gradient directions allows more accurate detection of diffusion variations along different directions [83,84], thus resolving multiple fiber orientations within a single voxel and improving the accuracy of tractography.

## Data preprocessing

*(i) Functional MRI data.* The functional images of all the children were preprocessed using SPM12 (https://www.fil.ion.ucl.ac.uk/spm) and DPABI 3.0 [85]. First, we removed the first 10 volumes and performed slice-timing correction. Next, we applied field map correction to reduce geometric distortion and realigned the volumes over time. After realignment, 94 rsfMRI scans were excluded due to excessive head motion with a criterion of maximum head motion > 3 mm or 3° or mean framewise displacement (mFD) [86] >0.5 mm. Then, the functional images were coregistered with individual T1 images and spatially normalized to a custom pediatric template using a unified segmentation algorithm [87] with the following steps: (i) individual structural images were initially segmented into 3 tissue (i.e., gray matter, white matter, and cerebrospinal fluid) probability maps by using the Chinese Pediatric Atlas (CHN-PD) (6 to 12 years) [88] as a reference; (ii) the resulting spatially normalized maps for each tissue type were averaged across scans to generate the custom tissue templates; (iii) individual structural images were segmented again using the custom tissue templates as a reference; and (iv) the functional images were spatially normalized using the transformation parameters estimated from the second segmentation of structural images. For the child cohort, custom tissue maps were used to improve the accuracy of spatial normalization. The normalized functional images were then resampled into 3-mm isotropic voxels and underwent spatial smoothing using a Gaussian kernel (full-width at half-maximum = 4 mm), linear detrending, and nuisance signal regression. A series of regressors were included in the nuisance regression, including 24 head motion parameters [89], "bad" time points with FD above 0.5 mm, white matter signals, cerebrospinal fluid signals, and global brain signals. Finally, we performed temporally band-pass filtering (0.01 to 0.1 Hz) on the images. For the adult cohort, the preprocessing procedures were conducted in the same way as for the child cohort, except that the functional images were normalized to the Montreal Neurological Institute (MNI) standard space.

*(ii) T1-weighted structural MRI data.* The T1-weighted structural images were preprocessed using FreeSurfer v6.0 [90]. First, we performed intensity normalization and removed non-brain tissue using the HD-BET algorithm [91], during which the automatically extracted brain tissue maps replaced the default maps (i.e., "brainmask.mgz") in FreeSurfer to improve accuracy. Next, we conducted tissue segmentation and cortical reconstruction on individual T1 images. Notably, the longitudinal processing stream of FreeSurfer was selected to obtain robust and reliable morphological measurements [92]. A trained researcher visually inspected the cortical reconstruction results to ensure that the correct boundaries were estimated. Finally, we obtained 5 local gray matter morphological features, including cortical volume, thickness, curvature, folding index, and surface area for each vertex.

*(iii) Diffusion MRI data.* The diffusion images were preprocessed using MRtrix 3.0 [93], FSL 6.0.1 (https://fsl.fmrib.ox.ac.uk/fsl/fslwiki/FSL), and ANTs [94]. First, the diffusion data were denoised, and Gibbs ringing artifacts were removed. Next, we corrected the eddy current-induced distortions, subject movement, and signal dropout for each scan using the FSL eddy tool [95]. Notably, 45 rsfMRI scans were further excluded from the structure–function association analysis due to missing images, excessive head motion (maximal motion >3 mm), or considerable signal dropout in the corresponding diffusion images. Then, field map correction was employed to reduce susceptibility by using FSL (epi_reg script). Finally, B1 field inhomogeneity was corrected with the N4 algorithm [96].

## Identification of the overlapping modular architecture in functional networks

We identified the overlapping modular architecture in the brain functional networks by employing an edge-centric module detection algorithm [30,44] (Fig 1B). Briefly, we first constructed a traditional functional network comprising nodal regions and interregional connectivities (i.e., edges). Then, we constructed the corresponding weighted edge graph representing the similarity between edges. Finally, we identified the module affiliations of each nodal region according to the module assignments of its edges in the corresponding edge graph.

*Functional network construction.* We constructed a brain functional network comprising 232 nodal regions for each rsfMRI scan of each participant. In this functional network, cortical nodes were defined based on a recently developed functional parcellation comprising 200 cortical regions (i.e., Schaefer-200) [97], and subcortical nodes were defined according to a subcortical functional parcellation comprising 32 regions [49]. We extracted the mean time series for each nodal region and estimated Pearson's correlation coefficients between every pair of nodes. Then, a weighted functional network was obtained by thresholding the correlation matrix with a density of 15% (i.e., 4,020 edges) to exclude potential spurious or weak correlations [98]. Negative correlations were not considered due to their controversial physiological interpretations [99,100].

*Edge graph construction.* For each functional network, we constructed a corresponding weighted edge graph that denoted the similarity between edges (Fig 1B). For simplicity and computation efficiency, we considered only directly connected edge pairs that shared one common node based on the hypothesis proposed in Ahn and colleagues [30], which captures the direct and dominant interaction between edges. The similarity between edges without common nodes was assumed to be zero. The Tanimoto coefficient [30,46] was used to incorporate the edge weight information. For a pair of edges $e_{ik}$ and $e_{jk}$ that share a common node $k$, their similarity was defined based on the similarity in the connection profiles between nodes $i$ and $j$:

$$S\left(e_{ik}, e_{jk}\right) = \frac{a_i \cdot a_j}{|a_i|^2 + |a_j|^2 - a_i \cdot a_j}, \tag{1}$$

where $a_i \cdot a_j$ represents the dot product of the 2 vectors $a_i$ and $a_j$, and $|a_i|^2$ denotes the sum of the squared weights of all connections of node $i$, and $a_i = \left(\tilde{A}_{i1}, \tilde{A}_{i2}, \cdots, \tilde{A}_{iN}\right)$ represents a modified connectivity profile of node $i$. Specifically, each element $\tilde{A}_{ij}$ is defined as follows:

$$\tilde{A}_{ij} = \begin{cases} A_{ij} & i \neq j \\ \dfrac{\sum_{m=1}^{N} A_{im}}{k_i} & i = j \end{cases}, \tag{2}$$

where $A_{ij}$ is the functional connection strength between nodes $i$ and $j$ and $k_i$ is the number of edges of node $i$. Of note, nonzero diagonal elements were included in Eq (2) to make the similarity definition, i.e., Eq (1), applicable to extreme cases in which nodes $i$ and $j$ are directly connected.

## Identifying the overlapping modular architecture based on the edge-centric module detection

Given a functional network of interest, we first detected the modular structure in the corresponding edge graph and then determined the module affiliations of each nodal region

according to the module affiliations of its edges. Here, the Louvain algorithm [47] was employed to detect the modular architecture in the large-scale edge graph, and each edge was assigned to a specific module. Each nodal region was then assigned to one or more modules due to diverse module affiliations of its edges, leading to spatial overlap between functional modules. A nodal region whose edges were involved in 2 or more modules was defined as an overlapping region. Notably, the detected module partition in the edge graph varied slightly across each instance of detection due to the heuristic property of the Louvain algorithm. The module number of the functional network was defined as the module number that appeared most frequently among 100 instances, and the other measurements regarding the overlapping modular structure were taken as the average across 100 instances of identification.

### Topography analysis of the overlapping modular architecture at the group level

To illustrate spatial patterns of the overlapping modular structure at different ages, we detected the modular architecture at the group level. All the children's rsfMRI scans were divided into 8 subgroups with a one-year interval (i.e., 6 to 7 years, 7 to 8 years, 8 to 9 years, 9 to 10 years, 10 to 11 years, 11 to 12 years, 12 to 13 years, and 13 to 14 years), and the adult group was set as the ninth subgroup for comparison. A group-level weighted functional network was constructed for each subgroup by averaging individual functional correlation matrices followed by network thresholding (i.e., density = 15%). Then, we detected the modular structure in the corresponding edge graph for each subgroup with 100 instances of module detection. To obtain a stable module division, we conducted the following 2 steps: (i) computed the module co-occurrence matrix between each pair of edges [101], wherein each element denoted the proportion of instances in which a pair of edges were assigned to the same module; and (ii) applied the modular detection algorithm to the module co-occurrence matrix 100 times. We iterated these 2 steps until the module partition remained unchanged across multiple detection instances. The final version of the module partition of edges was used to infer the overlapping modular architecture of the group-level functional network.

Considering the spatial overlap of functional modules, we obtained a spatial map for each functional module separately. Nodal values in the map denote the proportion of edges assigned to this module. To compare the modules observed in each child subgroup with those in adults, we further matched the functional module maps between the adult subgroup and all the child subgroups. Given a module map of interest in the adult group, a matching module map was selected as the map showing the maximal similarity for each child subgroup. The spatial similarity between 2 maps was calculated as the Pearson's correlation coefficient across nodal regions.

To further assess the functional system dependence of the overlapping modular structure, we mapped each cortical or subcortical node to one of the prior functional systems, including the visual, somatomotor, dorsal attention, ventral attention, limbic, frontoparietal, and default-mode systems [48] and the subcortical area [49]. For each module, we quantified its spatial pattern by estimating the percentage of nodes distributed in each functional system.

### Measure of the extent of module overlap in brain regions

Given an overlapping modular architecture, a node may be involved in multiple modules due to the diverse module affiliations of its edges. Here, a measure of entropy was used to quantify the extent of module overlap for brain regions by examining the distribution of module affiliations of all edges attached to a node (Fig 1B). Given a node $i$, the entropy [34] was defined as

follows:

$$H_i = -\sum_{k=1}^{n} p_{ik} \log_2 p_{ik}, \tag{3}$$

where $n$ is the number of modules involved with this node, $p_{ik}$ represents the proportion of its edges participating in module $k$, and $\sum_{k=1}^{n} p_{ik} = 1$. Nodal entropy was further normalized to a range of 0 to 1 by dividing it by $\log_2 n$ [34]. A higher normalized entropy value indicates greater overlap, suggesting that the distribution of edges among different modules is more homogeneous and diverse. For each scan of each participant, we obtained a nodal module overlap map in terms of nodal entropy.

To illustrate the patterns of nodal overlap at different ages, a group-level entropy map was separately generated for each child subgroup and the adult group by averaging individual entropy maps within the subgroup. Then, we estimated the spatial similarity of the entropy maps between each of the child subgroups and the adult subgroup by calculating Pearson's correlation coefficients across nodal regions. We further employed two-way repeated analysis of variance (ANOVA) models to explore whether the nodal overlap level was functional system-dependent for these subgroups (i.e., 8 child subgroups and the adult cohort). The functional systems were defined based on the prior 7 functional systems [48] and the subcortical area [49].

## Analysis of developmental changes in the overlapping modular architecture

To explore the developmental changes in the overlapping modular architecture, we assessed age effects on a series of brain measures, including the module number, the modularity of the edge graph, and the nodal module overlap (i.e., nodal entropy) at the global, system, and nodal levels. For each brain measure of interest, the age effects were estimated by using a mixed effects model [53,54]. The mixed model is suitable for a longitudinal data set with irregular intervals between measurements and is applicable to cases of missing time points. The model parameters were estimated with the maximum likelihood method. Considering the potential linear and quadratic age effects, we employed both the linear model and the quadratic model of the age effects. The optimal model was selected for each brain measure based on the Akaike information criterion (AIC) [102]. Sex and the in-scanner head motion parameter (i.e., mFD) were included as covariates within each model.

The linear model was defined as follows:

$$y_{ij} = \beta_0 + b_i + (\beta_{age} + b_{age,i})age_{ij} + \beta_{sex}sex_i + \beta_{mFD}mFD_{ij} + \varepsilon_{ij}. \tag{4}$$

The quadratic model was defined as follows:

$$\begin{aligned} y_{ij} &= \beta_0 + b_i + (\beta_{age1} + b_{age,i1})age_{ij} + (\beta_{age2} + b_{age,i2})age_{ij}^2 \\ &+ \beta_{sex}sex_i + \beta_{mFD}mFD_{ij} + \varepsilon_{ij}. \end{aligned} \tag{5}$$

In these equations, $y_{ij}$ represents the brain measure of subject $i$ at the $j$th scan, $b_i$ represents individual-specific intercept, $\beta_{age}$ represents the fixed effect, $b_{age,i}$ represents the random effect, and $\varepsilon_{ij}$ represents the residual. The fixed effect $\beta_{age}$ accounts for the population-level age effects, while the random effect $b_{age,i}$ accounts for individual variability in the age-related slope. The addition of the random effect allows the model to capture individual differences in developmental trajectories, thus providing a more accurate estimation of developmental changes. The fixed age effect was used to capture the developmental changes in the brain

measure. For the nodal-level analyses, the significance level of the results was corrected for multiple comparisons across nodes using the false discovery rate (FDR) method [103].

## Associations between cognitive functions and developmental changes in nodal overlap

To explore the cognitive significance of developmental changes in nodal overlap, we performed a meta-analysis using the NeuroSynth database (www.neurosynth.org) [55]. We first sorted the developmental changes in nodal entropy (i.e., age-related *t* values) in decreasing order and then divided the brain map into 10 bins (i.e., 10% apart as a box). Each bin was binarized to generate a brain mask. Next, we calculated the Pearson's correlations between each mask and all the cognitive term maps available in the database. We selected the top 2 associated cognitive terms for each mask and removed 6 overlapping terms among the 10 masks. Finally, 14 cognitive terms were used to depict the distribution of cognitive functions across different levels of developmental changes. The significance level of each spatial similarity was assessed using permutation tests that corrected for spatial autocorrelation [56]. Specifically, given a cognitive term of interest, we constructed a null distribution of correlation coefficients for each percentile bin separately based on 10,000 permutations. For each permutation, we generated a surrogate map preserving the spatial autocorrelation of the original age effect map, and then calculated the association with that specific cognitive term for each percentile bin. We further compared each empirically observed value to the corresponding null distribution to derive the significance level.

## Predicting chronological age from spatial patterns of nodal overlap

We used the SVR model with 10-fold cross-validation to test whether individual spatial patterns of nodal overlap could be used to predict a participant's chronological age. The nodal entropy map was set as features for each participant. To avoid the possibility of data leakage that could occur by including longitudinal scans from the same children in both the training and test sets, we performed the 10-fold cross-validation using 305 rsfMRI scans that were selected from independent subjects. For children with longitudinal scans, only 1 scan was selected from each individual and the other scans were discarded. To reduce the potential bias, the 10-fold cross-validation analysis was repeated 1,000 times with random scan selection. The detailed prediction process for each time of 305 randomly selected scans is as follows: First, all the entropy maps were divided into 10 subsets with similar age distributions. This division strategy is better than the random splitting method, which can reduce the sampling bias among the subsets [104,105]. Of the 10 subsets, one was designated as the testing set, and the remaining 9 subsets were used as the training set. Next, we linearly scaled each feature across individuals between zero and one in the training set and applied the estimated scaling parameters to the testing set. Then, we trained the prediction model for individual chronological ages based on the training set. Finally, we quantified the prediction accuracy by calculating the Pearson's correlation coefficients between the predicted scores (i.e., ages) obtained from the SVR model and the actual scores (i.e., chronological age). To assess the statistical significance of the prediction accuracy, we generated a null distribution of accuracy based on permutation tests ($n = 10,000$) by shuffling the actual scores across scans. To reduce the influence of confounding factors, we further corrected for sex, in-scanner head motion (i.e., mFD) and random age effects from the nodal entropy values prior to the prediction analysis. To determine the contribution of nodal features to the prediction model, we trained another SVR model using all the 305 scans to improve the estimation accuracy [104,105]. The resulting regression coefficients were regarded as the weights denoting the importance of all features. The final

contribution weights of nodal regions were obtained by averaging the estimated weights over the 1,000 times of random sampling. To further clarify the system dependence of the weights, we also classified the nodal weights into 8 functional systems, including 7 cortical functional systems [48] and the subcortical area [49]. The positive and negative weights were separately averaged within each system. Here, the SVR model was implemented using the LIBSVM toolbox in MATLAB with the default parameters (https://www.csie.ntu.edu.tw/~cjlin/libsvm/) [106].

## Analysis of nodal microstructure in the white matter structural network

In addition to local morphological measures, we also considered the potential influence of nodal microstructural properties in the white matter structural network. Structural networks were generated from the preprocessed diffusion images using DSI Studio (http://dsi-studio.labsolver.org). First, we tracked the fasciculus and obtained diffusion anisotropy parameters (i.e., FA). Specifically, we employed the generalized q-sampling imaging (GQI) algorithm [107] with a diffusion sampling length ratio of 1.25 for deterministic tractography. The Otsu threshold was 0.6. The tracking procedure was terminated if the turning angle was >45˚ or if the fibers reached the borders of the cerebrospinal fluid or subcortical areas. Ten million streamlines were generated with a step size of 0.625 mm, and only tracts with a length between 6~250 mm were retained for subsequent analysis. Next, the reconstructed streamlines were projected to the Schaefer-200 atlas, which has been registered to the native space. Finally, we calculated the FA values between every pair of nodes (i.e., the mean FA values along all reconstructed streamlines between 2 nodes) and further summed the FA values between each node and all the other nodes. This metric was defined as nodal FA strength, which indicates the microstructural property of the fiber bundles attached to a node [108].

## Predicting individual nodal overlap maps from structural features

To evaluate whether the spatial pattern of the nodal overlap map was associated with anatomical architecture, we performed a prediction analysis based on the SVR model for each scan of each participant. We considered 5 morphological measurements (cortical volume, thickness, curvature, folding index, and surface area) in the gray matter and the FA strength in the white matter. For each node, morphological features were extracted as the average value within the nodal region based on the Schaefer-200 atlas, which has been registered to the native space. Prior to the prediction analysis, we first examined the spatial similarity between nodal entropy and each structural feature of interest using a univariate association analysis (i.e., Pearson's correlation). Then, we performed the prediction using the same framework as that used in the age prediction mentioned above by integrating multiple structural features. Ten-fold cross-validation was also used here. In each validation instance, 10% of the nodal regions were designated as the testing set, and the remaining nodes were set as the training set. We predicted nodal entropy values for each scan by integrating anatomical features from both gray matter and white matter. To assess the statistical significance of the prediction accuracy, we generated a null distribution of prediction accuracy for the whole population using permutation tests. Specifically, we first generated a null distribution of accuracy by shuffling nodal entropy across nodes for each scan ($n = 100$), and then aggregated the null distributions across 446 scans, resulting in the final null distribution with 44,600 (i.e., 446 scans × 100 times) instances. The 95% significance level was determined according to the null distribution containing all 44,600 instances. To assess the prediction contribution of anatomical features, we trained another SVR model for each scan using all the nodal regions in the whole brain to improve estimation

accuracy [104,105]. The resulting regression coefficients were regarded as the weights denoting the importance of all features.

### Validation analyses

To ensure the robustness of the topography of the overlapping modular architecture, we investigated the potential influence of several network construction and analysis strategies in the adult cohort. Specifically, we examined the potential influence of functional parcellation for the nodal definition, network thresholding strategy, edge module detection algorithm, and nodal overlap estimation. First, we considered the spatial resolution of the functional parcellation. For the main analysis, we constructed a brain functional network comprising 200 cortical regions [97] and 32 subcortical regions [49]. To further validate the influence of spatial resolution, we reconstructed whole-brain functional networks, during which the cortical nodes were defined based on the Scheafer-100 atlas, which comprises 100 cortical regions [97]. Second, we examined the network thresholding density by obtaining weighted functional networks using 2 other network densities (i.e., 10% and 20%). Third, we explored the influence of the module detection algorithm. In addition to the Louvain algorithm [47], we employed the eigenspectral analysis method [109] to detect the modular structure of the edge graph. Finally, we quantified the nodal overlap level by considering the number of modules involved, which provides a more intuitive understanding. For each nodal region, we calculated the involved module number as the number of modules involved in its edges. The larger the involved number, the higher the nodal overlap in module affiliations.

To assess the robustness of the developmental effects of node entropy, we examined the potential impact of the network thresholding strategy, the statistical model for estimating age effects, and the head motion control strategy. First, as some weaker connections in functional networks play a key role in the development of human cognitive function [110], we examined age effects with a wider range of network thresholds, including 10%, 20%, and 30%. The latter threshold doubled the density of connecting edges used in the main results. Second, we used the GAMM [111] to assess the potentially complex nonlinear age effects. This model was implemented in R using the package mgcv. Third, to assess the potential effect of head motion, we performed a more stringent head motion control by excluding rsfMRI scans with mFD > 0.2 mm. Age effects were assessed using the mixed effects model based on the remaining scans.

### Declaration of competing interests

The authors declare that they have no known competing financial interests or personal relationships that could have appeared to influence the work reported in this paper.

### Supporting information

**S1 Fig. Univariate relationship in spatial patterns between nodal entropy and structural brain features within each scan in children.** For each scan, we separately calculated Pearson's correlation in spatial patterns between nodal entropy and each structural brain feature. **(A)** Frequency polygon of correlation coefficients for cortical thickness across rsfMRI scans. Compared to the null distribution, cortical thickness showed significant positive spatial correlations with nodal entropy in 17% of scans (74/446) ($p < 0.05$). **(B)** Frequency polygon of correlation coefficients for surface area in all rsfMRI scans. Compared to the null distribution, surface area showed significant negative correlations with nodal entropy in 44% of scans (196/446) ($p < 0.05$). **(C)** Frequency polygon of correlation coefficients for folding index in all rsfMRI scans. Compared to the null distribution, folding index showed significant negative

correlations with nodal entropy in 46% of scans (207/446) ($p < 0.05$). **(D)** Frequency polygon of correlation coefficients for FA strength in all rsfMRI scans. Compared to the null distribution, FA strength showed significant negative correlations with nodal entropy in 5% of scans (24/446) ($p < 0.05$). **(E)** Frequency polygon of correlation coefficients for cortical curvature in all rsfMRI scans. Compared to the null distribution, cortical curvature showed significant negative correlations with nodal entropy in 30% of scans (136/446) ($p < 0.05$). **(F)** Frequency polygon of correlation coefficients for cortical volume in all rsfMRI scans. Compared to the null distribution, cortical volume showed significant negative correlations with nodal entropy in 17% of scans (76/446) ($p < 0.05$). In (A–F), the inset in the upper corner denotes the null distribution of the correlation coefficients. This null distribution was generated by aggregating the 100 permutation instances for each scan, resulting in a total of 44,600 permutation instances (446 scans × 100 times). For each permutation within each scan, a surrogate nodal entropy map was generated that preserved the spatial autocorrelation characteristics of the original nodal entropy map [56]. The red line denotes the 95% significance level in the null distribution.
(TIF)

**S2 Fig. Spatial patterns of nodal module overlap (i.e., nodal entropy) at different spatial resolutions.** For the adult cohort, the overlapping modular architecture was separately detected in the group-level functional networks obtained from different functional parcellations. **(A)** Nodal overlap in the functional networks with a coarse parcellation. This network comprised 100 cortical nodes obtained from the Schaefer-100 atlas [97] and 32 subcortical regions [49]. **(B)** Nodal overlap in the functional networks with a fine parcellation (i.e., main results). This network comprised 200 cortical nodes obtained from the Schaefer-200 atlas [97] and 32 subcortical regions [49]. **(C)** Extent of nodal overlap for 8 systems at different spatial resolutions. Similar distributions of nodal overlap were observed between the 2 parcellations. VIS, visual; SM, somatomotor; DA, dorsal attention; VA, ventral attention; LIM, limbic; FP, frontoparietal; DM, default-mode; SUB, subcortical.
(TIF)

**S3 Fig. Spatial patterns of nodal module overlap (i.e., nodal entropy) for different network analysis strategies and their relationships with the main results.** All of these analyses were specified to the group-level functional network for the adult cohort. **(A)** Network density of 10% for functional network construction. **(B)** Network density of 20% for functional network construction. **(C)** Eigenspectral analysis for module detection in the edge graph. **(D)** Number of involved modules was used to quantify the extent of node module overlap. **(E)** Main result as a reference. In each case, all the network construction and analysis strategies were set to be the same as those in the main analysis, except for the strategy of interest. All correlations were assessed with Pearson's correlation across nodal regions. The significance of spatial similarity was assessed by comparing the observed value to a null distribution generated through 10,000 permutations that retained the spatial autocorrelation characteristics of the nodal entropy map of the original adult cohort in the main results.
(TIF)

**S4 Fig. Spatial patterns of developmental changes in nodal entropy for different thresholding strategies.** For each rsfMRI scan of children, the brain functional network was generated with different network thresholding strategies. **(A)** Network density of 10% for functional network construction. **(B)** Network density of 20% for functional network construction. **(C)** Network density of 30% for functional network construction. **(D)** Network density of 15% (i.e., main result as a reference). In each case, all the network construction and analysis strategies

were set to be the same as those in the main analysis, except for the strategy of interest. All correlations were assessed using Pearson's correlation across nodal regions. The significance of spatial similarity was assessed by comparing the observed value to a null distribution generated through 10,000 permutations that retained the spatial autocorrelation characteristics of the original age effect map ($t$-value map) in the main results.
(TIF)

**S5 Fig. Spatial patterns of developmental changes in nodal entropy using different statistical models. (A)** Age effects on nodal entropy using a mixed effects model. **(B)** Age effects on nodal entropy using a generalized additive mixed effects model. **(C)** Spatial correlation of age effects on nodal entropy between 2 statistical methods. The significance of spatial similarity was assessed by comparing the observed value to a null distribution generated through 10,000 permutations that maintained the spatial autocorrelation characteristics of the original age effect map (i.e., $p$-value map) obtained from the MEM in the main results. MEM, mixed effects model; GAMM, generalized additive mixed model.
(TIF)

**S6 Fig. Effects of head motion parameters on developmental changes of nodal entropy. (A)** Age effects on head motion parameters. Pearson's correlation analysis revealed that individual age and mFD values showed a significant but weak correlation. **(B)** Left: Age effects on nodal entropy. Right: Head motion effects on nodal entropy. The influence of age and head motion on nodal entropy was evaluated using a mixed effects model. In this model, age was designated as the independent variable, while the head motion parameter (i.e., mFD) was included as a covariate. The variable of sex was also included in the model as a covariate. **(C)** Left: Spatial pattern of age effects on nodal module overlap with a head motion control strategy of mFD < 0.5 mm (main results). Middle: Spatial pattern of age effects on nodal module overlap with a more strict head motion control strategy with mFD < 0.2 mm. Right: Spatial correlation of age effects on nodal entropy between 2 head motion control strategies. The significance of the spatial similarity was assessed by comparing the observed value to a null distribution generated through 10,000 permutations that retained the spatial autocorrelation characteristics of the original age effect map ($t$-value map) in the main results. mFD, mean framewise displacement.
(TIF)

## Acknowledgments

We thank the National Center for Protein Sciences at Peking University in Beijing, China for assistance on data acquisition.

## Author Contributions

**Conceptualization:** Tianyuan Lei, Xuhong Liao, Yong He.

**Data curation:** Tianyuan Lei, Xiaodan Chen, Weiwei Men, Yanpei Wang, Leilei Ma, Ningyu Liu, Jing Lu, Gai Zhao, Yuyin Ding, Yao Deng, Jiali Wang, Rui Chen, Haibo Zhang, Shuping Tan, Jia-Hong Gao, Shaozheng Qin, Sha Tao, Qi Dong, Yong He.

**Formal analysis:** Tianyuan Lei, Xuhong Liao.

**Funding acquisition:** Xuhong Liao, Sha Tao, Qi Dong, Yong He.

**Investigation:** Tianyuan Lei, Xinyuan Liang, Lianglong Sun.

**Methodology:** Tianyuan Lei, Xuhong Liao, Xinyuan Liang, Lianglong Sun, Mingrui Xia, Yunman Xia, Tengda Zhao, Xiaodan Chen.

**Project administration:** Xuhong Liao, Yong He.

**Supervision:** Xuhong Liao, Yong He.

**Visualization:** Tianyuan Lei.

**Writing – original draft:** Tianyuan Lei, Xuhong Liao, Yong He.

**Writing – review & editing:** Tianyuan Lei, Xuhong Liao, Mingrui Xia, Tengda Zhao, Yong He.

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
