## [Editor Report · Decision Letter 0]

2 May 2024

Dear Dr He, 

Thank you for submitting your manuscript entitled "Development of the overlapping network modules in the human brain" for consideration as a Research Article by PLOS Biology.

Your manuscript has now been evaluated by the PLOS Biology editorial staff as well as by an academic editor with relevant expertise and I am writing to let you know that we would like to send your submission out for external peer review.

Once your full submission is complete, your paper will undergo a series of checks in preparation for peer review. After your manuscript has passed the checks it will be sent out for review. To provide the metadata for your submission, please Login to Editorial Manager (https://www.editorialmanager.com/pbiology) within two working days, i.e. by May 04 2024 11:59PM.

Kind regards,

Christian

Christian Schnell, PhD

Senior Editor

PLOS Biology

cschnell@plos.org

---

## [Decision Letter · Decision Letter 1]

6 Jun 2024

Dear Dr He,

Thank you for your patience while your manuscript "Development of the overlapping network modules in the human brain" was peer-reviewed at PLOS Biology. It has now been evaluated by the PLOS Biology editors, an Academic Editor with relevant expertise, and by several independent reviewers. 

In light of the reviews, which you will find at the end of this email, we would like to invite you to revise the work to thoroughly address the reviewers' reports.

As you will see below, the reviewers are overall interested in your study and identified many strengths. However, there are also a couple of concerns, some of which represent quite a methodological challenge. In addition to the other concerns, we think that it is also important to address the comments on the claim of novelty, while demonstrating substantial advance by the current findings.

Given the extent of revision needed, we cannot make a decision about publication until we have seen the revised manuscript and your response to the reviewers' comments. Your revised manuscript is likely to be sent for further evaluation by all or a subset of the reviewers.

**IMPORTANT - SUBMITTING YOUR REVISION**

*Re-submission Checklist*

*Published Peer Review*

*PLOS Data Policy*

*Blot and Gel Data Policy*

Sincerely,

Christian

Christian Schnell, PhD

Senior Editor

PLOS Biology

cschnell@plos.org

REVIEWS:

Reviewer #1 (Adam Pines): This is a nice paper with several beneficial contributions. The description of the need for studying overlap in networks is very well-articulated, as are the methods. The longitudinal design of the study is another advantage, and the publicly available code is a clear strength.

I do have several comments, which are listed below. I'm suggesting major revisions, but I think they are minor as far as major revisions go.

Major Comments:

1. Figure 5 is interesting, but these correlations are quite low, particularly for brain maps. Please use something like you did for figure 6 (i.e., Burt et al 2020 or the spin test for parcels) to correct brain map associations by accounting for spatial autocorrelation. It's not clear that any of these associations are above chance, after accounting for spatial autocorrelation.

2. Please clarify if entire subjects are held out of train-test in figure 6. It is essential to not include the same subjects (across different observations) across train-test splits to avoid leakage.

3. The multimodal and cognitive aspects are good efforts but are a bit oversold. They might be interesting enough to warrant inclusion as either a main text or supplemental figure, but I don't think its fair to say the authors have "provided novel insights into the neural mechanism underlying individual cognitive development.", especially because the structure-function association is multivariate rather than tractable to a more straightforward univariate correspondence. What are the mechanisms, and furthermore, where are the individual-level measures of cognition? Please clarify or temper these claims.

4. Maximum head motion > 3 mm or 3° or mean framewise displacement (mFD) > 0.5 mm: seems like an incredibly liberal threshold. Could the authors provide a surface/parcel plot of entropy associations with FD parallel to their surface/parcel plot of entropy associations with age? I would imagine there are a ton of residual motion effects, and it would be useful for the reader to know whether or not this is the case, particularly if they can compare effect sizes/prominence relative to those reported for age.

Minor Comments:

1. The authors appear to make two claims of primacy that are unsubstantiated. Despite citing work from the same lab, I assume the authors were unaware of our study on overlapping functional modular organization in children and adolescents when they claimed "our study is the first to uncover overlapping functional modular organization in children and adolescents" (See Pines et al., Nature comms, 2022). We also take a personalized approach, directly tie individual level cognitive effects in, developed a generalized estimating equations approach to run stats on individual locations belonging to multiple networks, and explicitly quantify network edge development as a function of the hierarchical similarity (hierarchical distance) as alluded to by the current authors in this manuscript's discussion. You are not obligated to cite us, but the claim of primacy is false and should be removed. Further, although not in developing individuals, this paper is similar enough to "Edge-Community Entropy is a Novel Neural Correlate of Aging and Moderator of Fluid Cognition", now out in J Neurosci, where that work probably should be cited. The second unsubstantiated claim of primacy, although indirect, is "However, until now, there has been no gold standard for evaluating the quality of the detected overlapping functional modular structure in the human brain." I'm not sure the authors meant it this way, but it sounds like they are implying that they pioneered an approach to evaluate the quality of overlapping functional modular structure in the human brain. Unless I missed something substantial in the paper, this appears to be taking credit for the development of the approach, which by my understanding was done by the Betzel lab (Faskowitz et al), not the current authors. Apologies if I'm misunderstanding this claim, but I would at least clarify it.

2. Reliance on linear models might be one reason the authors are not returning a ton of significant results in figure 3. There's every reason to believe development is non-linear in this age range, as many of the authors' citations point to (see citation 9 for a particularly compelling example). I think the results they returned are real, but choice of linear mixed effect models seems like a suboptimal fit to the data at hand compared to something like generalized additive mixed effect models. It's not 100% necessary for this paper to make a contribution to the literature, but seems like a clear step forward.

3. I would recommend describing what went up and down in the sentences: "Statistical analysis revealed that seven nodal regions showed significant linear changes with age (pFDR corrected < 0.0014, Figure 3C). These regions showed dissociable age-related changes, with significant increases mainly located in the superior and inferior parietal lobules and the lateral prefrontal cortex and significant decreases mainly located in the ventral and medial prefrontal cortex and the putamen." I assume it's entropy but that's never actually stated in results or the figure caption.

4. Could the authors allocate a sentence or two to describing why they are fitting age as a random effect rather than purely as a fixed effect with random effects for "subname"?

5. I might avoid using the phrase "brain age", as this paper does not seem to be about discrepancies between predicted and actual age (which is a good thing!). Brain age literature is rife with problems (see Pitfalls in brain age analyses. Human brain mapping) that this current paper is not subject to.

6. It's not clear that you aren't using topological and topographical interchangeably. It's largely a stylistic choice to be consistent with that or not, and clarifying or standardizing your usage of these words is not something that will change my accept/non-accept opinion. However, some believe these refer to entirely different constructs (topology being purely in graph-space, topography representing a spatial distribution).

7. This is also just my opinion and not something I would suggest rejecting this paper for, but I don't think this reasoning is entirely coherent: "The high nodal overlap in the sensorimotor regions indicates the functional diversity of this system, which is consistent with its involvement in multiple modules (Figure 2E). In addition to the widely known sensorimotor functions [54], these regions may be involved in other cognitive functions or behaviors, such as attention allocation [55], speech perception [56], and emotional regulation". In many papers, including the Margulies PNAS paper cited several times, there is a clear case for sensorimotor regions being the most specialized for one specific modality out of all cortical regions (i.e., unimodal). The authors are by no means obligated to take the same explanation we have in similar papers (Pines et al., Nature comms, 2022, Luo et al., Nature comms, 2024), but as is I find their logic inconsistent with the citations they invoke.

8. Edge-centric analyses are an important attempt to get at different aspects of network organization, but are subject to important limitations. Please consider the work "A mathematical perspective on edge-centric brain functional connectivity" in nature comms in either your introduction or discussion.

9. I can't find a protocol paper or more detail on it, but it's odd that a HARDI scan would have ordinary diffusion weightings only. Would the authors mind confirming that only b=0 and b=1,000 were used? Or citing a protocol paper if one exists?

10. "Then, a weighted functional network was obtained by thresholding the correlation matrix with a density 5 of 15% (i.e., 4020 edges) to exclude potential spurious or weak correlations." This seems antagonistic to the idea of analyzing overlapping associations, which as the authors show, often fall into secondary or tertiary "network memberships" or more broadly FC strengths. Given that the data is already heavily reduced to 232 parcels, there doesn't seem to be any computational need for this step. Can the authors justify this choice? I do appreciate the inclusion of S2A and S2B to show similarity in the relative (although not absolute) distribution of entropy across regions across thresholds.

11. "To assess the statistical significance of the prediction accuracy, we generated a null distribution of accuracy using permutation tests (n = 100) by shuffling the actual entropy across nodes for each scan." Is n=100 correct? Looks like 10k in main text, and the p value for 7b indicates 10k permutations (because it can't be more precise than p<0.01 with 100 tests). If I'm simply misunderstanding, it would be great if the authors could clarify this.

12. Further, it's not clear that leave-region-out is a valid approach due to spatial autocorrelation. Please use leave entire-subjects-out if you want to include this analysis, or explain to me why that is not possible/valid in this instance. In my opinion, univariate associations between entropy and structure metrics are more informative than multivariate regression here, particularly if the authors are trying to claim something mechanistic.

Reviewer #2: In this work, the authors use a large longitudinal sample of children ages 6-14 years of age to look at how overlapping module assignment in edge-centric functional networks varies with age. They find that areas that show high and low overlap in assignment are relatively stable with age, with higher overlap in VA and SM networks and lower in VIS and DM. Some particular regions show age-related changes, such that their module assignment becomes more (PFC) or less (parietal cortex) stable with age. The authors also conduct ML analyses using SVR to show that module assignment overlap can predict age, and that regional variation in module overlap is associated most strongly with regional variation in patterns of cortical thickness.

The study has several strengths, including a large longitudinal sample and novel methods, and I very much appreciate the set of validation analyses. There are some weaknesses, specifically around clarity and interpretation of findings as well as methodological assumptions, that could be remedied. I've included specific comments on these aspects below.

Major comments

Intro is well-motivated for a study focusing on soft partitioning of functional networks and overlap of network assignments…but needs more motivation/explanation as to why taking an edge-centric approach in addition to a soft partitioning approach is necessary. It seems the authors are taking two steps from what has previously been done, rather than the one that they motivate in the intro, which makes comparison to previous studies harder.

Similarly, in the discussion, the authors frame their findings in the context of previous work on overlapping functional modular organization, but don't mention how their methods differ significantly (in the focus on edges) from other prior work that used nodal FC but characterized overlapping modular architecture.

There has been some work using soft partitioning methods that capture overlap in the age range studied that is not covered in the intro. Tooley et al 2022 (Neuroimage) and Faskowitz et al. 2018 (Sci Reports) come to mind. Dworetsky et al. 2021 (Neuroimage) and Hermosillo 2024 (Nat Neuro) might also be relevant, though in adults.

The reader needs clearer explanation of what goes into constructing a weighted edge graph, and perhaps an addition to Fig 1. Given that an edge represents the similarity of fluctuations in BOLD signal between two regions, what does the weighted edge graph represent? What is being evaluated for similarity between edges? I had to go look at Faskowitz 2020 to remind myself.

Thresholding at an arbitrary density is problematic for interpretation of results. Other work using edge-centric methods does not necessitate a specific threshold or the removal of negative edges before constructing an weighted edge matrix. This simply indicates that all results are likely based on only examining the strongest connections, while there is good evidence that weak connections might be changing during development and play an important role. The authors should conduct supplemental analyses using fully weighted matrices and/or matrices at a wider variety of densities to investigate this.

"For simplicity, we considered directly connected edges that shared at least one common node, and the similarity between edges without common nodes was assumed to be zero. "

This seems like an extremely strong assumption, and not one that I'm familiar with in other work. No rationale is provided or supplemental analyses showing that this is a reasonable assumption to make are included. 

Is the Tanimoto coefficient a typical way to assess edge similarity? I see no mention of it in Faskowitz et al (2020). Or is this paper also introducing a different way of assessing edge similarity than has been previously used? If so, why?

If the main paper finding is related to the nodal entropy map (which the validation analyses suggest is the case), I'd suggest emphasizing the interpretation of this specific finding in the abstract and first paragraph of discussion in a clearer way. Findings that there are decreases in nodal overlap in module affiliations in PFC imply what for brain development? To my mind, it implies that PFC is becoming more homogenously affiliated, or more stable/less flexible in its network affiliation, with age. Some phrases of this nature would be helpful for a reader not versed in soft-partitioning methods.

Minor comments

Some clearer takeaways in the abstract might be appreciate by readers (e.g. what does regional inhomogeneity in module overlap mean for children's brain networks?)

The values of edge correlations in Fig 2a seem quite low, compared to the corresponding figure in Faskowitz 2020, and this figure further reinforces that setting many edge similarities to 0 without calculating them may be a strong assumption.

Would Fig 1d be better off as a graph with differently colored or weighted lines? It is initially difficult to interpret what the reader is trying to understand from the colors and sizes of circles.

It is quite interesting that looking at Fig 3b and comparing to adults, it seems that the overall trend might be decreasing entropy with age (both in insular areas and subcortex); this would imply that module assignments based on edge similarity are more stable in adults than children. Did the authors examine average whole-brain entropy (or cortical vs subcortical) associations with age?

The claim on page 13 line 27 seems quite strong, given some of the other research that exists already.

I'd find the title easier to parse as "Development of overlapping network modules in the human brain". I would be more specific than "signatures" when describing associations with brain structure.

---

## [Decision Letter · Decision Letter 2]

19 Aug 2024

Dear Dr He,

Thank you for your patience while we considered your revised manuscript "Development of overlapping network modules in the human brain" for publication as a Research Article at PLOS Biology. This revised version of your manuscript has been evaluated by the PLOS Biology editors, the Academic Editor and one of the original reviewers.

Based on the reviews and on our Academic Editor's assessment of your revision, we are likely to accept this manuscript for publication, provided you satisfactorily address the remaining points raised by the reviewers (specifically, Reviewer 1's comment regarding the legend of Figure 6B). Please also make sure to address the following data and other policy-related requests:

* We would like to suggest a different title to improve readability: "Functional network modules overlap and are linked to interindividual connectome differences during human brain development"

* Please include information in the Methods section whether the study has been conducted according to the principles expressed in the Declaration of Helsinki.

* Can you please check that the following blurb is correct: "Network modules in the human brain have mostly been considered to be non-overlapping during development. This neuroimaging study of functional connectome maturation during childhood and adolescence in over 300 children reveals its overlapping architecture, which is associated with structural properties."

* DATA POLICY:

Regardless of the method selected, please ensure that you provide the individual numerical values that underlie the summary data displayed in the following figure panels as they are essential for readers to assess your analysis and to reproduce it: 4A, 6C, and 7B.

* CODE POLICY

We expect to receive your revised manuscript within two weeks. 

*Published Peer Review History*

*Press*

Sincerely,

Christian

Christian Schnell, PhD

Senior Editor

cschnell@plos.org

PLOS Biology

Reviewer remarks:

Reviewer #1: The authors demonstrated a strong effort in addressing our comments. I am convinced of their findings and commend their work. I also appreciate them clarifying and explaining several points to me.

I'm suggesting acceptance, but here are two remaining minor points for the authors to consider.

1. The caption for Figure 6B isn't clear. The gray frequency polygon is referenced, but not the blue one. I assume the first sentence after (B) refers to blue, but simply adding an equivalent sentence structure for the blue frequency polygon would make the difference between what they are intended to display more clear.

2. I'm not a fan of displaying only p-values from models as they have done in figure S5. Because it's a supplementary analysis and because GAMMs don't have straightforward coefficients, it's not a deal-breaker as is. But I would avoid doing that in future work.

Best of luck to the authors moving forward.

---

## [Editor Report · Decision Letter 3]

29 Aug 2024

Dear Dr He,

Thank you for the submission of your revised Research Article "Functional network modules overlap and are linked to interindividual connectome differences during human brain development" for publication in PLOS Biology. On behalf of my colleagues and the Academic Editor, Claus Hilgetag, I am pleased to say that we can in principle accept your manuscript for publication, provided you address any remaining formatting and reporting issues. These will be detailed in an email you should receive within 2-3 business days from our colleagues in the journal operations team; no action is required from you until then. Please note that we will not be able to formally accept your manuscript and schedule it for publication until you have completed any requested changes.

PRESS

Sincerely, 

Christian

Christian Schnell, PhD

Senior Editor

PLOS Biology

cschnell@plos.org